# Building Virtual 3D City Model for Smart Cities Applications: A Case Study on Campus Area of the University of Novi Sad

**Dušan Jovanović *** , **Stevan Milovanov, Igor Ruskovski, Miro Govedarica , Dubravka Sladić , Aleksandra Radulović and Vladimir Pajić**

Faculty of Technical Sciences, University of Novi Sad, 21000 Novi Sad, Serbia; s.milovanov@uns.ac.rs (S.M.); rus_igor@uns.ac.rs (I.R.); miro@uns.ac.rs (M.G.); dudab@uns.ac.rs (D.S.); sanjica@uns.ac.rs (A.R.); pajicv@uns.ac.rs (V.P.)

* Correspondence: dusanbuk@uns.ac.rs; Tel.: +381-63-103-8664

**Abstract:** The Smart Cities data and applications need to replicate, as faithfully as possible, the state of the city and to simulate possible alternative futures. In order to do this, the modelling of the city should cover all aspects of the city that are relevant to the problems that require smart solutions. In this context, 2D and 3D spatial data play a key role, in particular 3D city models. One of the methods for collecting data that can be used for developing such 3D city models is Light Detection and Ranging (LiDAR), a technology that has provided opportunities to generate large-scale 3D city models at relatively low cost. The collected data is further processed to obtain fully developed photorealistic virtual 3D city models. The goal of this research is to develop virtual 3D city model based on airborne LiDAR surveying and to analyze its applicability toward Smart Cities applications. It this paper, we present workflow that goes from data collection by LiDAR, through extract, transform, load (ETL) transformations and data processing to developing 3D virtual city model and finally discuss its future potential usage scenarios in various fields of application such as modern ICT-based urban planning and 3D cadaster. The results are presented on the case study of campus area of the University of Novi Sad.

**Keywords:** virtual 3D city models; CityGML; Smart Cities applications; LiDAR

## 1. Introduction

The concept of Smart City integrates information and communication technology (ICT) and various physical devices and sensors connected to the Internet of Things (IoT) network to optimize the efficiency of city services and better connect to citizens [1]. Technology that relates to Smart City allows city officials to interact with community and city infrastructure and to monitor what is happening in the city, how the city is evolving, and what actions are needed to improve the quality of city services. For that purpose, ICT is used. There are many Smart City applications developed for the purpose to model and manage urban flows and to allow the near real-time responses to challenges. However, the term itself remains unclear to its specifics and therefore open to many interpretations [2], and possible applications. Nevertheless, one thing is for certain and that is that Smart Cities involve the use of modern ICT technologies in all of the areas of application.

Considering that Smart Cities imply complex distributed systems which may involve multiple stakeholders, applications, sensors, and IoT devices, spatial data infrastructure can play an important role in establishing interoperability between systems and platforms, in order to be able to link and use such heterogeneous data [3]. Based on the open and international standards of the Open Geospatial

Consortium (OGC) [4], it is possible to develop a spatial data infrastructure that integrates different sensors, IoT devices, simulation tools, and 3D city models within a common operational framework. For that purpose, the Open Geospatial Consortium has formed Smart Cities Domain Working Group [5] whose role is to implement OGC activities and standards within a Smart Cities environment to address issues such as energy efficiencies, pollution, waste handling and recycling, land and other resource use, and to ensure sustainability of resources for future generations. The aim is to enable interoperability between different applications, by using the shared data and processing models and to enable the smooth transitions and process flows that include monitoring, planning, execution of changes, and maintenance of the existing systems.

The Smart Cities data and applications need to replicate, as faithfully as possible, the state of the city and to simulate possible alternative futures. In order to do this, the modelling of the city should cover all aspects of the city that are relevant to the problems that require smart solutions. The important modelling aspects include: networks and flows that can describe all motion in the city; building and facilities models (historic, present and future), including all aspects that could impact the city, its environment, its citizens and their activities, capable to support planning and maintenance through 4D (3D and time) aspects of urban planning, cadaster, environmental factors, etc.

Smart cities are aimed to efficiently manage growing urbanization, energy consumption, maintain a green environment, improve the economic and living standards of their citizens, and raise the people's capabilities to efficiently use and adopt the modern information and communication technology (ICT). In the smart cities concept, ICT is playing a vital role in policy design, decision, implementation, and ultimate productive services [6].

Integration of all smart systems (such as smart home, smart parking, etc.,) and the IoT devices (such as sensors, actuators, and smartphones) in the city can play a vital role to develop the urban services by building their city digital and smarter. However, interconnection of lots of IoT objects to collect urban data over the Internet to launch a smart digital city affects vast volume of data generation, termed as Big Data. Thus, it is a challenging task to integrate IoT devices and smart systems in order to harvest and process such big amount of real-time city data in an effective manner aimed at creating a Smart Digital City [7]. At the core of smart city lies the sensors and actuators embedded in the smart devices that sense the environment for facilitating effective decision-making. This involves integration of several information and communication technologies like artificial intelligence, protocols, Internet of things (IoT), wireless sensor network (WSN) etc., [8]. Unfortunately, current accomplishments in smart city in Southeast Europe, are in most cases theoretical and not empirical. The most researched themes are smart governance and smart environment [9].

The need for Smart Cities standards within the OGC environment is categorized into four different areas: (1) data/spatial data, (2) policy and governance, (3) data sharing platforms, and (4) applications, services, and solutions. In this regard, one of the major goals is to collect relevant spatial and non-spatial data that cover the city area. For this purpose, OGC has developed a set of standards for representing 2D and 3D geospatial data that can support the development of the spatial data infrastructure on the city level for the Smart Cities applications. CityGML [10] and IndoorGML [11] are standards that deal with 3D outdoor and indoor information. Those are semantic models that can help build 3D city database which can be further expanded and linked to the datasets of other relevant organizations such as urbanism or cadaster. The CityGML model has been fully revised in preparation for the release of CityGML version 3.0 [12] to reflect the increasing need for better interoperability with other relevant standards in the field [13]. CityGML 3.0 will enable mapping of constructive elements from Building Information Modeling (BIM) data sets given in the IFC standard onto CityGML. In general, CityGML 3 is designed to better address needs of Smart Cities and IoT applications.

In terms of spatial location, data can be continuous or categorical. Continuous data are defined for each point of the geographical space of interest such as topography. Categorical data include objects with boundaries. They are represented by points, lines, and polygons. The spatial position of geographical objects is determined using two primary data acquisition methods. The first one is

field measurements, mainly land surveying, Global Navigation Satellite System (GNSS) positioning, and Light Detection and Ranging (LiDAR) [14].

One of the methods for collecting data that can be used for developing such 3D city models is Light Detection and Ranging (LiDAR). LiDAR is a modern remote sensing technology and it finds its applications within many different areas. One of them is 3D city modelling considering the fact that it can collect vast amounts of data in a short period of time. With advancements in airborne LiDAR technology, it is possible to model urban topography at an unprecedented spatial resolution and granularity and extract previously unavailable characteristics of individual buildings [15]. Furthermore, this technology has also provided opportunities to generate large-scale 3D city models at low cost [16]. The collected data are further processed to obtain fully developed photorealistic 3D city models. This process can be automated using a machine learning approaches such as point cloud classification methods [16,17]. Optical aerial imagery can be combined to a LiDAR point cloud for the generation of photorealistic city models [18].

Aim of this research is to develop virtual 3D city model as a basis for 3D spatial data infrastructure that will support Smart City applications and provide overall framework for integration of different datasets related to urban environment and city services, beyond mere 3D visualization. The aim is to develop a case study that is based on LiDAR surveying, point clouds, ETL transformations, and development of virtual 3D city models and 3D city database, that will allow users to employ these models for further analysis and visualization tasks in a variety of application domains such as urban planning, cadaster, facility management, environmental simulations, cultural heritage, indoor or outdoor navigation.

The paper is structured as follows. After the introduction, theoretical background and related work on 3D city models and CityGML standards, as well as their area of application have been given in Section 2. Section 3 describes the surveying methods used to develop 3D city model of the case study area (campus area of the University of Novi Sad, nearby Petrovaradin fortress and part of city center). Section 4 presents the results of such development and schema transformation. Discussion on possible applications and future developments of 3D city model is given in Section 5. In this discussion, the main focus is on the applications in the domain of urban planning and 3D cadasters, as one of the areas that can benefit a lot from such developments. However, these are by no means, the only areas where such models can be used. Conclusions are given afterwards.

## 2. Theoretical Background and Related Work

### 2.1. CityGML

CityGML is the international standard of the Open Geospatial Consortium (OGC) for the representation and exchange of 3D city models. It defines the three-dimensional geometry, topology, semantics, and appearance of the most relevant topographic objects in urban or regional contexts. These definitions are provided in different, well-defined levels-of-detail (multiresolution model).

Similar to more traditional 2D spatial data, 3D city models represent an approximation of the real world. Quantity and content of a city model is directly related to the usage of the 3D model. The amount of detail within a 3D model, in a sense of geometry and attributes is commonly known as level of detail (LoD), which describes how thoroughly the spatial object is modelled. LoD is a concept in GIS and 3D city modelling. LoD concept is important at every step of a 3D city model development, even before any data acquisition is performed [19]. According to Gröger et al. [20], a LoD0 model is simply a 2.5-dimensional digital terrain model, which is a two-dimensional map with a 3D terrain. LoD1 models are simple box models. LoD2 models add roof structure to LoD1 models. LoD3 models have detailed exterior features, such as openings and wall structures. LoD4 models are the most complex models with building interior features. However, LOD4, which is used for representing the interior of objects (such as indoor modelling of buildings) in CityGML 2.0, which is the current published standard at the moment of writing of this paper, is expected to be removed

from the announced CityGML 3.0. Instead, only the LODs 0/1/2/3 remain and the interior of objects is expressed integrated with the LODs 0/1/2/3 [13].

Furthermore, CityGML supports application domain extensions (ADE) in which users can create their own extensions for their particular applications. Different LoDs and application domain extensions (ADE) could broaden the application area of CityGML which has been demonstrated in many researches [21–27].

CityGML aims to define basic entities, attributes, and relations of 3D city models. CityGML defines city model object classes that are identified as required or relevant in many applications. CityGML also includes different LoDs which allow different concepts of buildings, GIS, and BIM to be joined into a comprehensive data model [28].

CityGML allows semantic and schematic interoperability of 3D city models within the spatial data infrastructure. It represents the base for data exchange between different geographic information systems with no data quality or quantity losses [10]. Interoperability aspects with other technical models such as LADM, INSPIRE, or IFC are of special interest of the forthcoming CityGML 3.0. In this regard, a space concept has been introduced in CityGML 3.0. A space is an entity of volumetric extent in the real world. Spaces are further subdivided into physical spaces and logical spaces. Physical spaces are spaces that are fully or partially bounded by physical objects, while logical spaces can also be bounded by non-physical, virtual boundaries and they can represent aggregations of physical spaces (e.g., a condominium as a logical space consists of several physical spaces—rooms). This space concept can be used as a basis for the integration with LADM and IndoorGML. The new construction module is defined for the purpose of mapping IFC elements onto CityGML classes. Dynamizer and versioning modules are used for modelling changes and city dynamics in order to manage time-dependent properties. The PointCloud module adds the possibility to use 3D point clouds to represent the geometries of physical spaces and space boundaries. Despite the introduction of the new modules, backward compatibility with CityGML 2.0 is sustained.

Indoor space differs from outdoor space in many aspects. Basic concepts, data models, and standards of spatial information should be redefined to meet the requirements of indoor spatial applications. The requirements of indoor spatial information are differently specified according to the types of applications. In general, the applications of indoor spatial information are classified into two categories. The first category is management of building components and indoor facilities and the second category deals with the cases of indoor space usage. Building construction management and facility management belong to the first category. While the main focus of the first category is on building components such as roofs and walls, the second category is focused on usage and localization of features (stationary or mobile) in indoor space. The indoor spatial information of the second category is to represent spatial components such as rooms and corridors, and constraints such as doors. For example, indoor location-based services, indoor route analysis, or indoor geo-tagging services belong to the second category. The aim of the IndoorGML standard [11] is to define a framework for spatial data about indoor space that can locate stationary or dynamic elements within the object and provide all relevant spatial services that deal with their position within the space instead of representing the architecture itself. However, a need for the integration of CityGML and IndoorGML has been shown in [29] for 3D building models combining both indoor and outdoor scenarios. As already mentioned, the forthcoming CityGML 3.0 anticipates a better integration with IndoorGML through the introduction of the space and space boundary concepts.

## 2.2. Applicability of 3D City Models

Gröger and Plümer (2012) [20] give an overview of CityGML, its underlying concepts and its LoDs, how to extend it through ADE, possible applications, and its potential future development. The focus of CityGML is on the semantical aspects of 3D city models allowing users to employ virtual 3D city models for advanced analysis and visualization tasks in a variety of application domains such as urban planning, 3D cadasters, environmental simulations, cultural heritage, facility management, etc. This is

in contrast to purely geometrical, graphical models such as KML or VRML which do not provide sufficient semantics. CityGML is based on the geography markup language (GML), which provides a standardized geometry model. Because of this model and its well-defined semantics and structure, CityGML supports interoperable data exchange in the context of geospatial web services and spatial data infrastructures. Therefore, the main goal of the CityGML-based 3D city models is not only to provide 3D visualization of the urban area, but also to store relevant city data and create city databases, which should serve as a basis for different application fields, and in general it can provide overall spatial data infrastructure for different Smart City applications.

According to the literature review one of the major areas of applications of the 3D city models for Smart Cities is related to modelling and simulations that have impact on the environment, such as energy efficiency of urban areas, urban building energy modelling and city-wide energy simulations [30–35], greenhouse gas inventories, $CO_2$ emission and low-carbon cities [36,37], urban microclimates analysis [38], solar energy resources in cities using 3D city models [39], urban energy simulation for climate protection [40], etc.

3D cadasters are also an important potential area of application of CityGML standard. It is the growing trend that cities produce the city models according to the CityGML, usually from the point clouds produced by LiDAR. Such data could be reused for 3D cadaster purposes. Several studies have addressed the analysis of the possibility of using CityGML to link legal and physical spaces, by the means of application domain extension (ADE), paying particular attention to the buildings [41,42]. 3D buildings at LoD2 may be created as a combination of 2D digital building ground plans derived from the official digital cadastral maps and LiDAR data [43]. A practical implementation of the CityGML ADE based on land administration domain model [44] has demonstrated the benefits of providing relations between spatial objects from the legal and physical world. An application domain extension to CityGML for immovable property taxation in Turkey has been proposed by Çağdaş [45]. A 3D spatial data model of the solar rights associated with individual residential properties has been proposed by Li et al. [46]. The model is extended from the land administration domain model and implemented by CityGML and it can be integrated to cadastral database. A CityGML-LADM application domain extension model is proposed for the ownership structure of condominium units by Li et al. [47]. The spatial structure of 3D data models in CityGML and industry foundation classes (IFC) were investigated in the management of 3D public law restrictions by Kitsakis et al. [48]. Integrating legal and physical dimensions of urban environments through utilization of BIM and GIS models such as CityGML and IFC were also addressed [49,50]. In Serbia, several researches have been conducted to prove the usefulness of 3D formats such as CityGML and IFC in the context of 3D cadaster [51,52].

In the domain of urban management and smart cities, Eicker et al. [53] proposed an integrated urban platform as the essential software infrastructure for smart, sustainable, and resilient city planning, operation, and maintenance, which is based on 3D city models and workflows for city scale building energy modelling. A method to support energy retrofitting of historic districts is proposed by Egusquiza et al. [54] and a CityGML is enriched with cultural and energy domain extensions. Chaturvedi et al., proposed access control in spatial data infrastructures for Smart Cities based on 3D city models and CityGML [3]. CityGML RESTful Web service for automatic retrieval of CityGML data based on their semantics was proposed by Pispidikis and Dimopolou [55]. This Web service is conceptually designed to achieve CityGML data retrieval based on their semantics characteristics and deliver data in JSON format in the context of 3D Web services.

## 3. Study Area and Methods

### 3.1. Study Area

City of Novi Sad (45°15′N, 19°50′E) is located in the northern part of Serbia, i.e., on the southern part of the Pannonian Plain and it is the second largest city in the country with a population of about 300,000 in a built-up area of approximately 130 km$^2$. The geographical area is plain, from 76 to 86 m

a.s.l., with a gentle relief. Besides the urban part of the city Novi Sad, the administrative area is divided by the river Danube and comprises of the city of Novi Sad and one small city Petrovaradin, three town and ten villages. Case study area consist of the part of city center, campus area of University of Novi Sad and Petrovaradin fortress, which is shown in Figure 1.

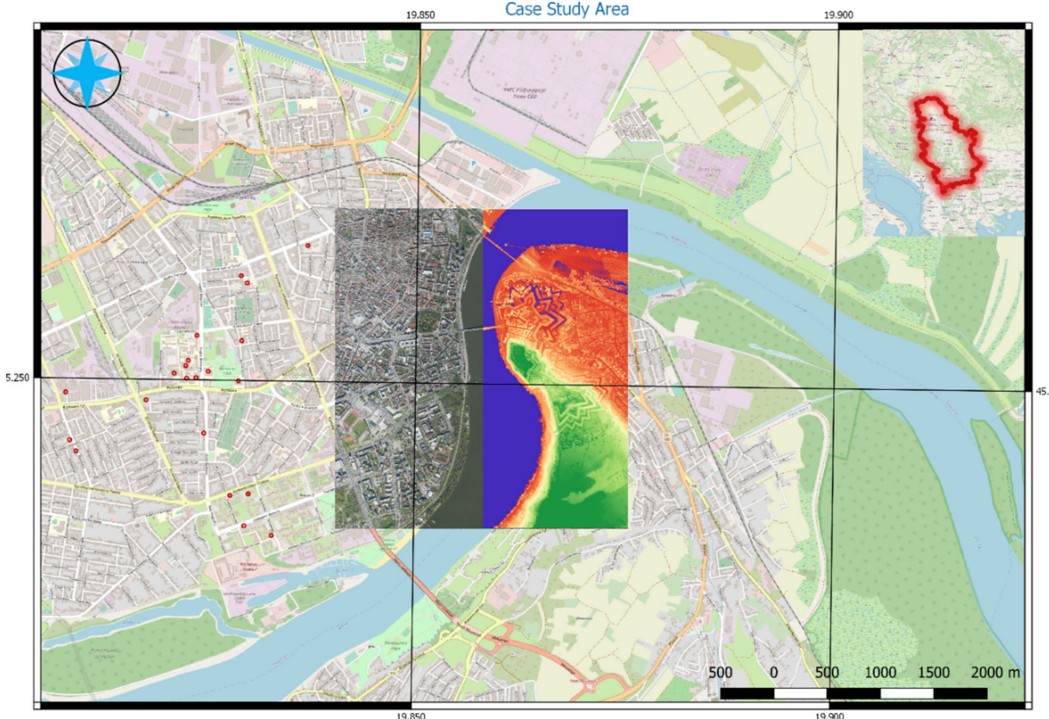

**Figure 1.** Case study area with DTM 0.5 m and Orthophoto 0.1 m/pix.

## 3.2. Acquisition Parameters and Basic Processing

For the massive data collection methods LiDAR platform was used. For this purpose, the Riegl LMS-Q680i laser scanner and digital camera DigiCAM H39 was used. All the data from the sensors were geo-codified owing to a direct georeferencing system consisting of a GPS-IMU navigation system, composed by a GPS receiver and an inertial sensor.

The flight altitude, according to the orography of the area and safety standards, has been maintained about 200 m AGL and speed about 45 km. The density of the raw laser pulses was of about 40 pt/sqm. The geodetic system of the survey was the ETRS89-UTM34N. The final products were delivered in the same system.

The georeferenced and calibrated flight lines are merged and processed to filter out the low points, aerial points, false echoes, etc. Because of physical, software, and hardware constraints, found also on the last generation computers, related to the management and elaboration of a finite number of points, the cloud of points was split in square blocks with a side of 250 m. The first classification was performed using quick automatic algorithms, performed in TerraScan (http://www.terrasolid.com/products/terrascanpage.php). Classification in TerraScan offers a number of classification routines that affect or rely on points or a group of points, or routines that can be used only in macros. During laser data processing the following classes are defined: ground, vegetation, low points, roofs of buildings, and default.

Ground routine classifies ground points by creating a triangulated surface model iteratively. There are two routines, one that is best suited for classifying ground in airborne laser data sets and in data sets where there is mainly natural terrain, and second that is best suited for classifying ground where there is mainly hard ground surface, such as paved roads or other areas. We have used both,

by drawing polygons around natural/paved areas in order to apply one inside the polygons and the other routine for the rest of the data.

Below surface routine classifies points that are lower than the neighboring points in the source class. This routine is run after ground classification, to locate points which are a bit below the true ground surface. For each point, the routine finds up to 25 closest neighbor points in the source class. It fits a planar or curved plane to the neighboring points and computes the average magnitude of the elevation differences between the points and the plane. This routine was used to classify low points.

By height from ground routine classifies points that are within a given height range compared to a reference surface. This routine is also used for classifying points into different vegetation classes for preparing, for example, building roofs, or tree detection. Buildings routine classifies points on building roofs that form a planar surface. The routine requires that ground points have been classified before. It is also advisable to classify points above the ground into a separate class, so that this class contains points in an elevation range above ground where building roofs are included. The routine starts from empty areas in the ground class and tries to find points on planar surfaces above these areas. Those two routines were used for automatic classification of vegetation and roofs of buildings. All other points were left in default class.

On the other hand, the image acquisition with mentioned camera above, allows to create digital orthophoto. The processing chain of the orthophoto making is the following:

- Download and backup of the raw images
- Camera trajectory output
- Raw data processing
- Frames georeferencing, mosaicking, and orthorectifying.

During the raw data processing, the single frames are converted in uncompressed TIFF format with a first radiometric balance and correction. Using the aerial trajectory, GPS data and the timestamp of the shots, the images are mosaiced, with a semi-automatic definition of the tie points and of the cutlines between overlapping images.

Airborne Lidar sensors provide dense height information of large areas in an efficient manner and it can be used for the development of the 3D city model [56]. In our case study it has been operated with the parameters of 40 points per square meter, and aerial images with 5 cm per pixel.

For the production of digital elevation model (DEM) from the point cloud, the filtering of the point cloud should be carried out in order to remove points representing surface of non-ground objects. For the ortho-rectification a digital terrain model (20cm/px) derived from the cloud points was used, filtering only the significant points of the ground. For this purpose, we have also used TerraScan, which use a filtering algorithm based on triangular irregular networks, which adapts to the data points from below and handles surfaces with discontinuities [57,58]. Because of its efficiency, in the last years we have seen a lot of examples of using Lidar data for process of ortho-rectification, not only for airborne images, but also for satellite images [59,60].

This chain processing assures a perfect correspondence among ortho-images, cloud points, and terrain model, considering the fact that the same trajectory is used for the data elaboration. After this process, the ortho-images were provided in a TIFF format for the visualization and analysis. The resolution of the ortho-images was 10 cm.

In this case study, after LiDAR scanning, as a first step orthophoto images and a georeferenced point cloud have been created. Before any further steps, this data had to be processed and prepared. Data preparation included the following steps: initial classification, detail classification, 2D or 3D vectorization, and creation of digital elevation model. Initial classification usually depends on the software solution, but all of them classified LiDAR data in several similar classes (default, ground, low vegetation, medium vegetation, high vegetation, building, low points, and model key points). As was mentioned before, all these steps were done in TerraScan, using already prepared batch macros for initial classification and for creation of digital elevation model. Detail classification is usually according

to the project specification, and in this case it was made in accordance with all available layers that were recognized as relevant for this case study, but also in relation to the existing legal regulations, which were later used for vectorization. After that, 2D and 3D vectorization was done using classified LiDAR data and according to the relevant rules in the Republic of Serbia [61]. List of all layers can be seen in Table 1. Finally, results of these basic processing steps were a fully classified point cloud, and a file containing 3D building models in a CAD structure, which is shown on Figure 2.

**Table 1.** 2D and 3D vector layers.

| 2D CAD/Vector Layers | 3D CAD/Vector Layers |
|---|---|
| Asphalt road, field road, pedestrian road, forest, park, isolated trees, bush, terrace, river, stream, water, canal, fence, railway, bridge, parking, bench, bus stops | House, building, residential and commercial building, auxiliary building, lighting, pole, power lines, traffic sign |

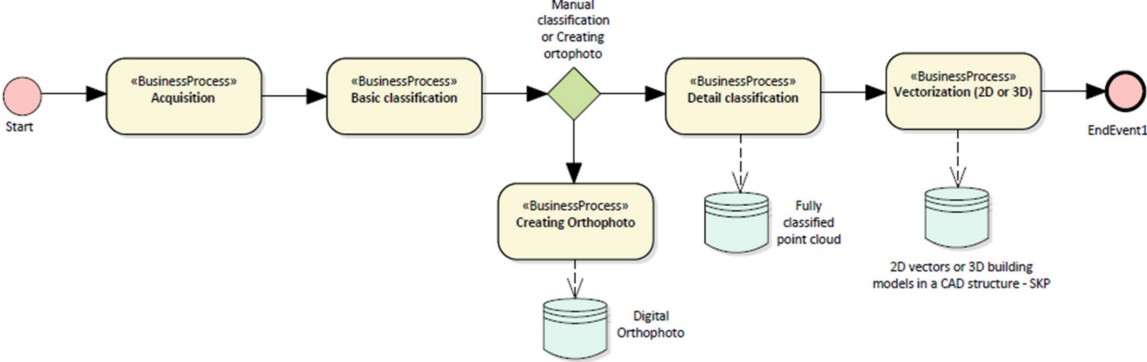

**Figure 2.** Basic processing steps.

## 4. Results—Development of Virtual 3D City Model

### 4.1. Data Transformation

After classification and 2D and 3D vectorization of point cloud, 3D building models in the resulting CAD structure can be used in multiple ways. The basic usage of this CAD structure is only for pure visualization, but for advanced and further usage in various applications it must be transformed into GIS or BIM formats. There are several steps that need to be taken in order to fully transform CAD format such as DGN or DXF to IFC or GML format. Figure 3 shows the proposed workflow of the data transformation process.

First solution is to create a CityGML model directly from a CAD structure, or a CAD file organized in layers (Figure 3—ETL Transformation 1). Using proposed ETL Transformation 1 it is possible to create model according to CityGML standard almost automated, because it skips the step of manually processing in a BIM tool. Part of this workflow is transformation from 3D CAD structure into GML model with or without roof texture.

On the other hand, as it can be seen on the Figure 3, for fully textured 3D model according to the CityGML standard, previously generated DGN file was imported to BIM tool software in order to add photorealistic textures to the 3D model. After that, as a one solution IFC format is available directly, or using ETL transformation 2 and provided algorithms and functions for manipulation of the 3D objects, we have appropriate textured 3D model according to the CityGML standard (Figure 4). For the model to comply with GIS standards, it is important that it stores geometric as well as topologic and semantic characteristics of the object. We use ETL transformation 2 that forms the CAD data, create fully designed models in one of the two most commonly used standards for storage, visualization, and most importantly, data analysis – CityGML and BIM/IFC (Figure 3).

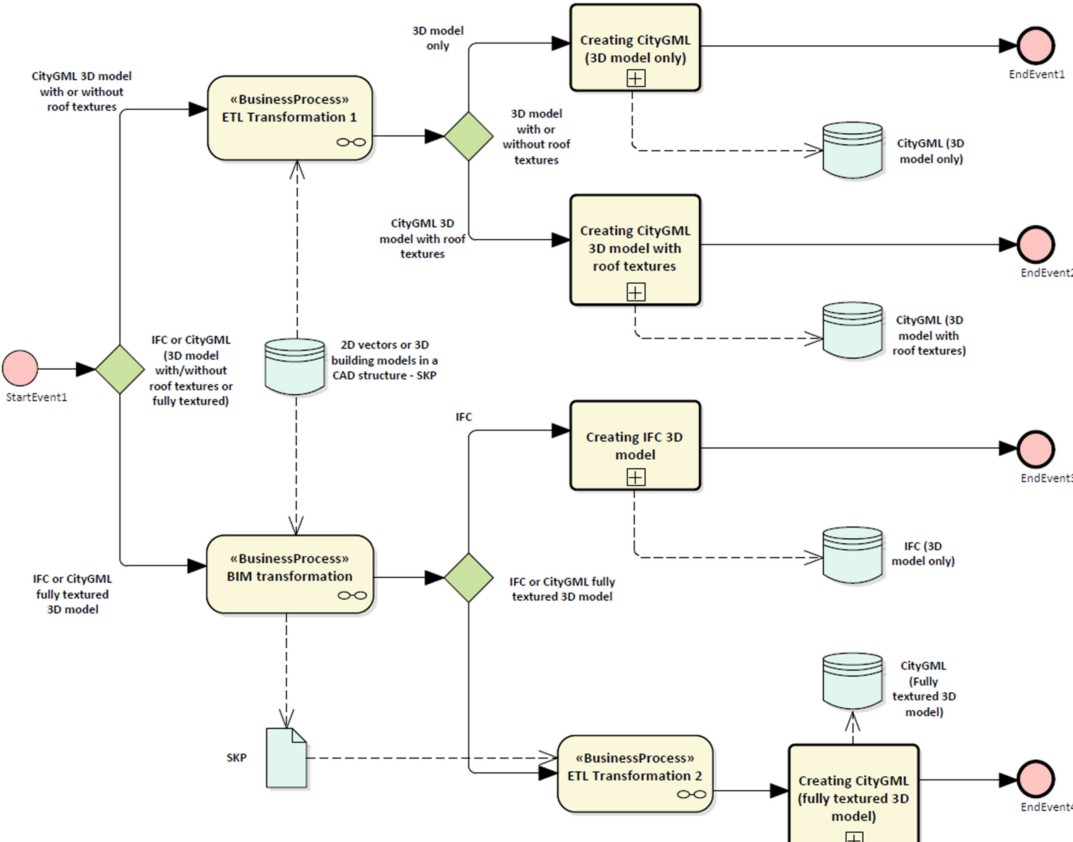

**Figure 3.** The defined workflow of the data transformation process.

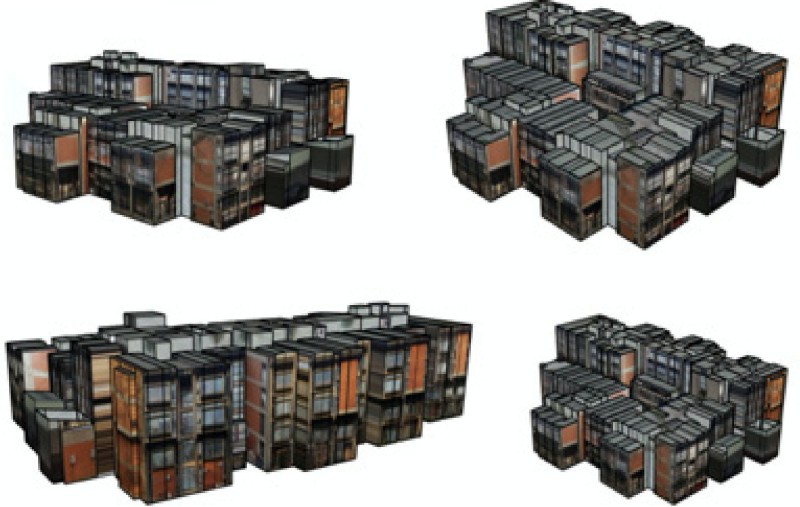

**Figure 4.** 3D model of the building of the Faculty of Philosophy and Law of the University of Novi Sad in the campus area.

As it can be seen in Figure 3, in order to transform SketchUp (SKP) model to CityGML standard, it must be decomposed and transformed into several components defined by the standard, which is done by ETL Transformation 2. Workflow of this transformation from to a GML format is shown in Figure 5.

Through this decomposition and transformation, all the important semantic attributes are added, and the photorealistic model in GML format, was created using FME tool (https://www.safe.com/fme/). This transformation schema is shown in Figure 6 and consists of several key parts.

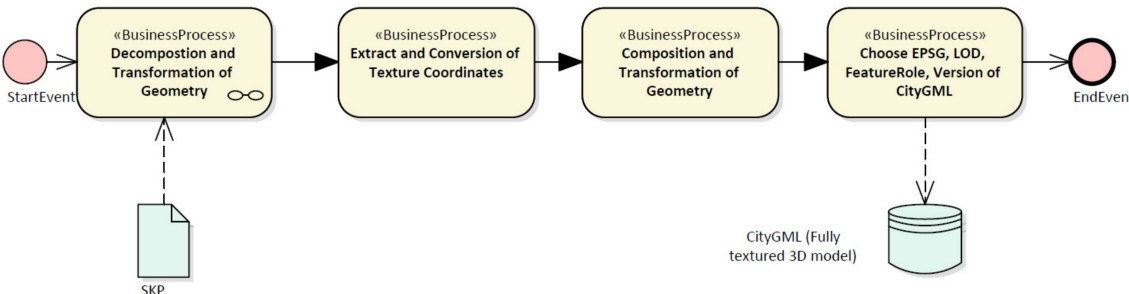

**Figure 5.** Workflow from SKP to a geography markup language (GML) format transformation.

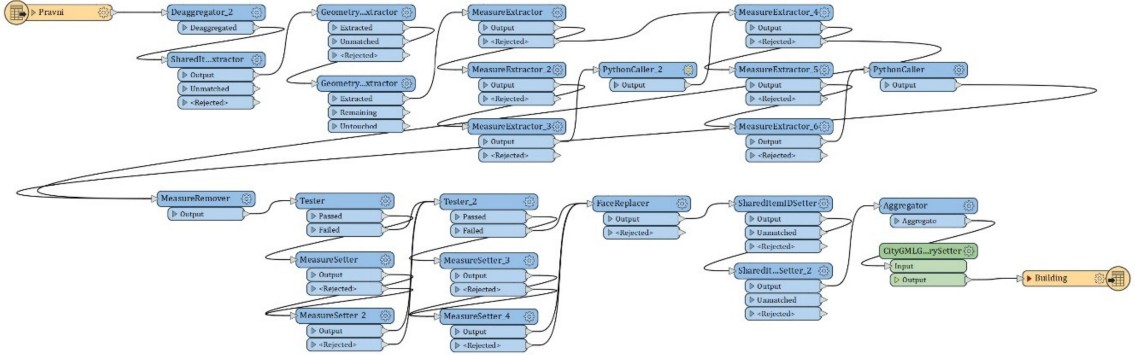

**Figure 6.** Scheme of ETL transformation 2 for SKP photorealistic model created in BIM tool into a GML format.

The first part of this transformation is the part that relates to the transformation of the texture coordinates from input SKP model, into the texture coordinates that are known in the CityGML standard, respectively. This if followed by a section dealing with the decomposition and translation of SKP model geometries (geometry collection) into surface geometries that are understandable to CityGML. After this step, each individual geometry (geometry surface), is assigned to appropriate texture coordinates, or photorealistic images that are CityGML compliant. The final part of ETL transformation, is to consolidate all surface geometries into one entity that represents a single object, and to assign appropriate CityGML properties with the help of the CityGML Geometry Setter transformer. This transformer allows to add properties such as CityGML LoD name, lod3MultiSurface, as well as CityGML feature roles, or City object member. Also, in the last step, the coordinate reference system in which the object of interest is located is defined. In this transformation, as well as in all other ETL transformations that are used in this case study, EPSG: 32634, Serbian national CRS is used. The final result of the transformation from Figure 6 is the CityGML model of a single object (Figure 7a), a feature type element Building, a LoD 3 with a photorealistic representation in the corresponding coordinate system.

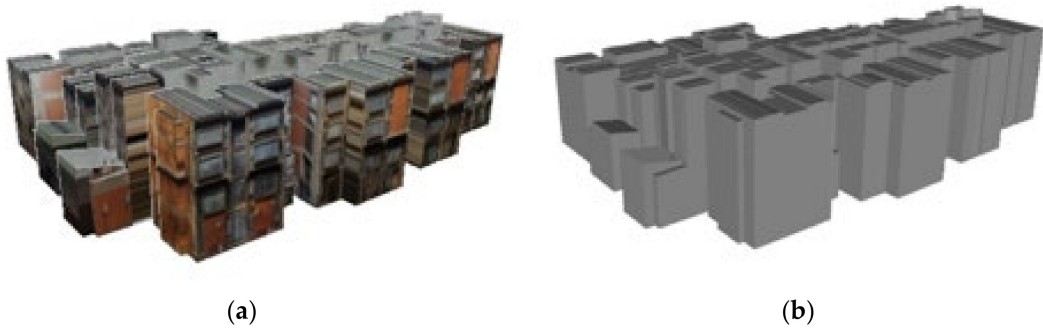

(**a**)　　　　　　　　　　　　　　　　　　　　　　　　(**b**)

**Figure 7.** 3D model in GML format (**a**) and IFC format (**b**).

The same process was applied for the SKP to IFC translation. IFC components are IfcBuilding, IfcBuildingElementProxy, IfcBuildingStorey, IfcProject, and IfcSite that represent spatial structure of the building [52]. Among these five components, only IfcBuildingElementProxy is geometric, while the rest are attribute components. Figure 7 represents the model of the same object in GML (Figure 7a) and IFC (Figure 7b) format.

Another way to create a CityGML model is to create it directly from a CAD structure, or a CAD file organized in layers (Figure 3 – ETL Transformation 1). This method of creating model according to CityGML standard is much more automated because it skips the step of manually processing in a BIM tool. Workflow for transformation from 3D CAD structure into GML model with or without roof texture is shown in Figure 8.

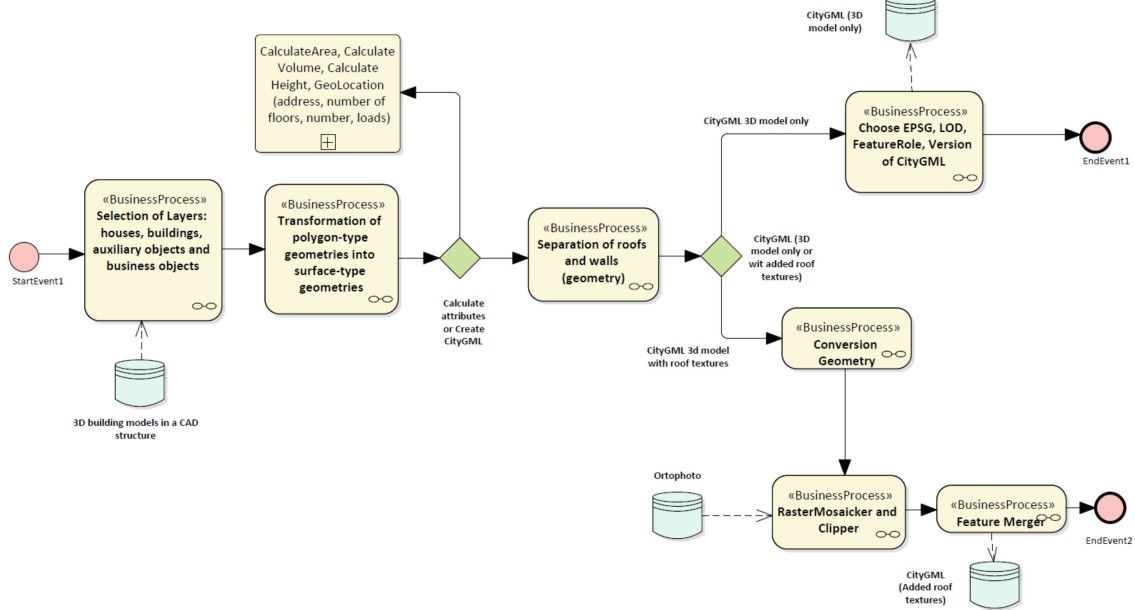

**Figure 8.** Transformation from 3D CAD structure into GML model with or without roof texture.

This ETL transformation is done in two steps. The first transformation (Figure 9) refers to the transformation of the CAD model into a GML model without textures, and the second transformation (Figure 10) to add texture of roofs from orthophoto images to object models that were previously created in GML format.

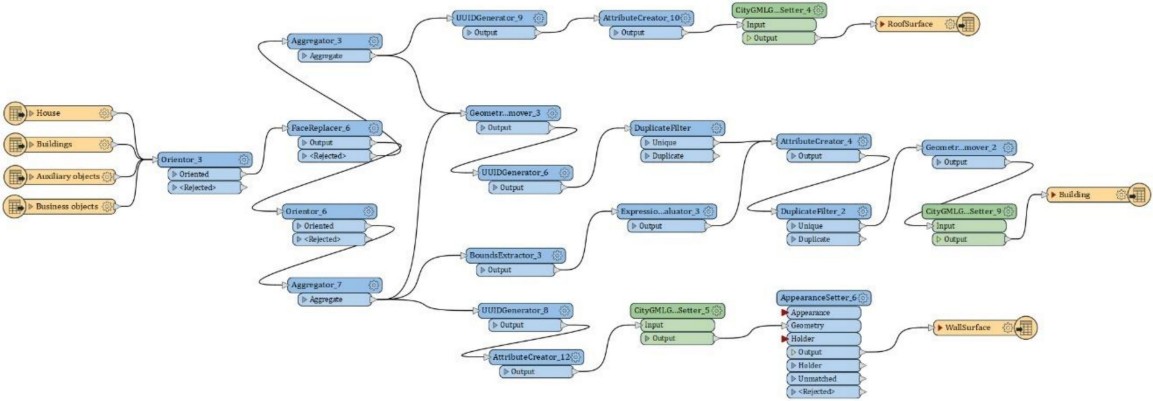

**Figure 9.** ETL transformation scheme for vectorized CAD models into GML model without textures.

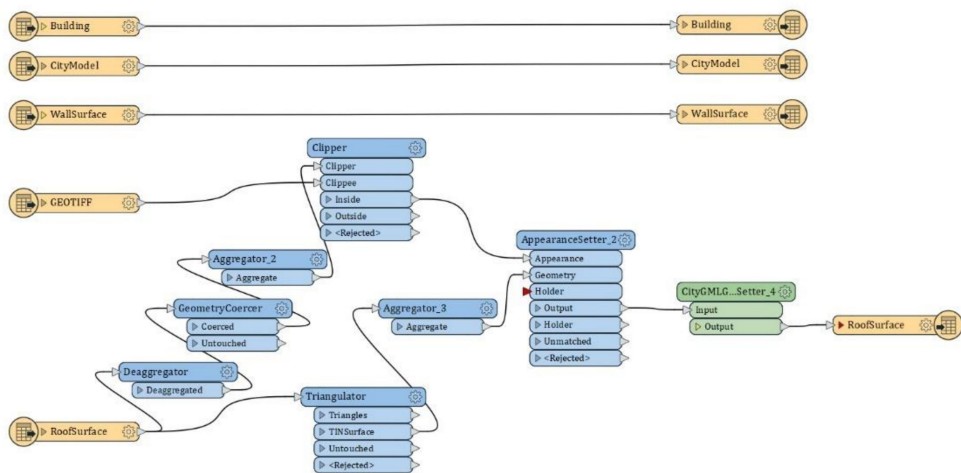

**Figure 10.** ETL transformation schema for adding textures of roofs from orthophoto images to building models that were previously created in CityGML standard.

First ETL transformation (Figure 9), at the beginning contains four input layers of CAD format: houses, buildings, auxiliary objects, and business objects. As in the previous ETL transformation, in order for the data to be transformed into GML format, it is necessary to transform them from polygon geometries into surface geometries, and orient them all in the same direction, more precisely by the left-hand rule. The orientation step performed at the start trough the orienter transformer, is important in order to have uniformly oriented surfaces for the future work of adding textures to specific side of a surface (roof or wall surfaces) as well as for distinguishing the exterior and the interior side of a surface when building solid geometries. As for the CityGML feature type WallSurface and RoofSurface elements, they carry geometry information for each object that must be preserved in ETL transformation, so that they can be used in future topological analyses in the Smart 3D city model. The CityGML feature type building is a purely attribute element, and in this case, it does not carry information about geometries, but it stores all the necessary information that is relevant for each of the buildings in the 3D model of the city. As an example of adding attributes to building type, real heights for each object are extracted from the coordinates of each object individually and added to the appropriate CityGML attribute defined by the standard itself (citygml_measured_height). Another attribute related to object heights are the units of measure in which the heights are expressed (citygml_measured_height_units). In addition to these attributes, the buildings type defines the level of detail of each building, feature roles, LoD name. Also, if there is information for each object such as the street where it is located, building number, number of floors, floor area, and others they are added as part of this feature type. All three feature types (Building, RoofSurface, and WallSurface) have common gml_id over which they are uniquely defined. Also, CityGML Geometry Setter transformers were assigned to all three feature types and coordinate systems were defined (EPSG: 32634). As a result of the ETL transformation from Figure 9, the CityGML model of 1204 of the city buildings (located in the city center, campus, and Petrovaradin Fortress) of LoD2 was developed. This model is set in the appropriate coordinate system and it does not contain photorealistic elements.

The problems that were noticed during the creation of the described ETL transformation, i.e., the 3D model of the city, are that some objects lost some of their geometric characteristics after the transformation. For example, some buildings did not get their roof geometries, or they got them in part, as well as the problem that the footprints of some of the buildings are not where they should be, but are about 80 m lower (at 0 m above sea level). Subsequent analysis of the aforementioned models of objects showed that the problem did not originate from the ETL transformation. The root of the problem was in the models created in CAD structure through automatic 3D building vectorization. Some of the buildings were not properly vectorized and it resulted in faulty geometries according to the CityGML standard. This problem was solved by a repeated 3D vectorization of selected objects.

The second transformation (Figure 10) is in fact a continuation of the transformation of already created buildings in the CityGML standard (Figure 9), but with the addition of the textures of the roofs from the orthophoto images, all according to the CityGML standard.

The purpose of this transformation is to clip previously mosaiced orthophoto images, with the edges of the roofs of the buildings, to get the perfect match and to satisfy the conditions of photorealism, above all the credibility and texture of surfaces, naturalness and radiation of such model. It is very important for this transformation to use orthophoto images that were created at the same time as the laser scanning because they are already in the same coordinate system and were created from the same platform (the problem of the offset orthophoto in relation to the building models did not occur). This ETL transformation is used to transform the CityGML surface geometries into polygon geometries so that the raster could be clipped. After that, gml_ids of the roofs were added to those raster's that represented the roofs and thus to each roof an appropriate roof image was assigned. As a result of the transformation from Figure 10, the CityGML model of 1204 of the city buildings (city center, campus and Petrovaradin Fortress) of LoD2 with photorealistic elements of the roofs and in an appropriate coordinate system were created. Some of the problems observed when creating this ETL transformation are that the images taken are not true orthophotos and this has been reflected as a problem on tall objects. Roof textures are not identically projected onto the very geometry of the roofs of tall buildings. Another of the observed shortcomings is that oblique images were not taken during the scan to create the textures of the walls of the buildings. This shortcoming is foreseen for future research that will be built on this research. Figure 11 shows a 3D model of the city in the Cesium Web Map Client with elements of the photorealistic roofing and the corresponding attributes for each of the buildings.

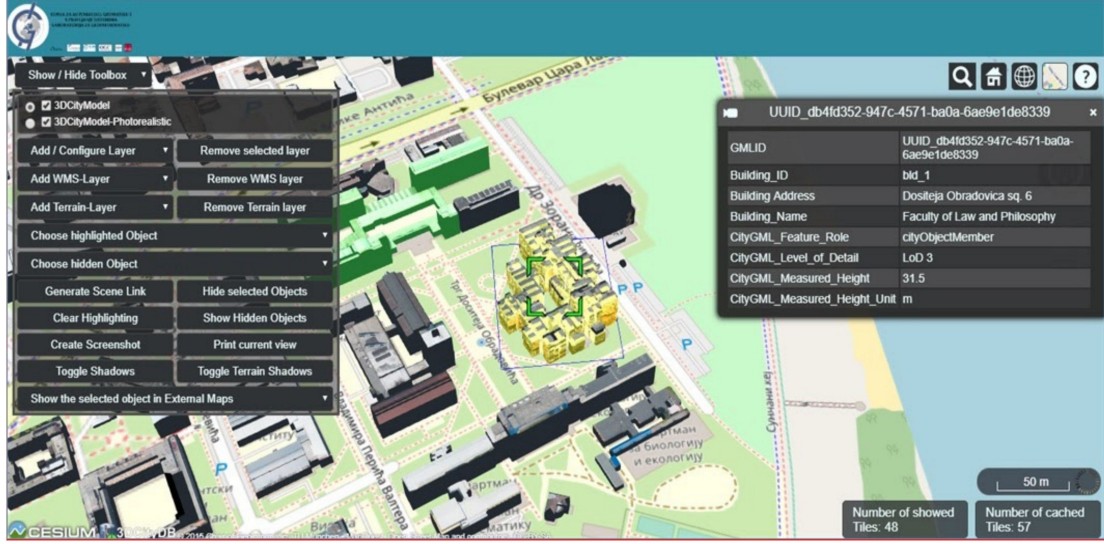

**Figure 11.** View of the 3D campus model in CityGML standard with associated attributes attached to each object in the Cesium JS Web Map Client (http://geoinformatika.uns.ac.rs/index.php/en/virtual-campus/).

One of the main advantages of mass LiDAR surveying and processing and automatic classification of point cloud data is results of classification and possibility to connect results with features defined by the CityGML standard. Table 2 gives an overview of all CAD layers, which have been transformed into GML format by ETL transformation, that is, into features defined by the CityGML standard. Figure 12 shows these features. After the 3D model has been created, all necessary validations of the obtained GML format have been performed to determine whether the obtained model meets all the geometric, topological, and semantic characteristics defined by the CityGML standard. The 3D city model thus created, in accordance with the CityGML standard, was imported into 3DCityDB Database (https://www.3dcitydb.org/3dcitydb/).

**Table 2.** CAD layers which have been transformed into 3D model defined by the CityGML standard.

| CAD Layers | CityGML Features |
|---|---|
| House, building, auxiliary object, residential and commercial object | Building, RoofSurface, WallSurface |
| Breaklines, grid | TINRelief |
| Forest, thicket | PlantCover |
| Road, field road, pedestrian road | Road |
| Parking, other | LandUse |
| Water | WaterBody |

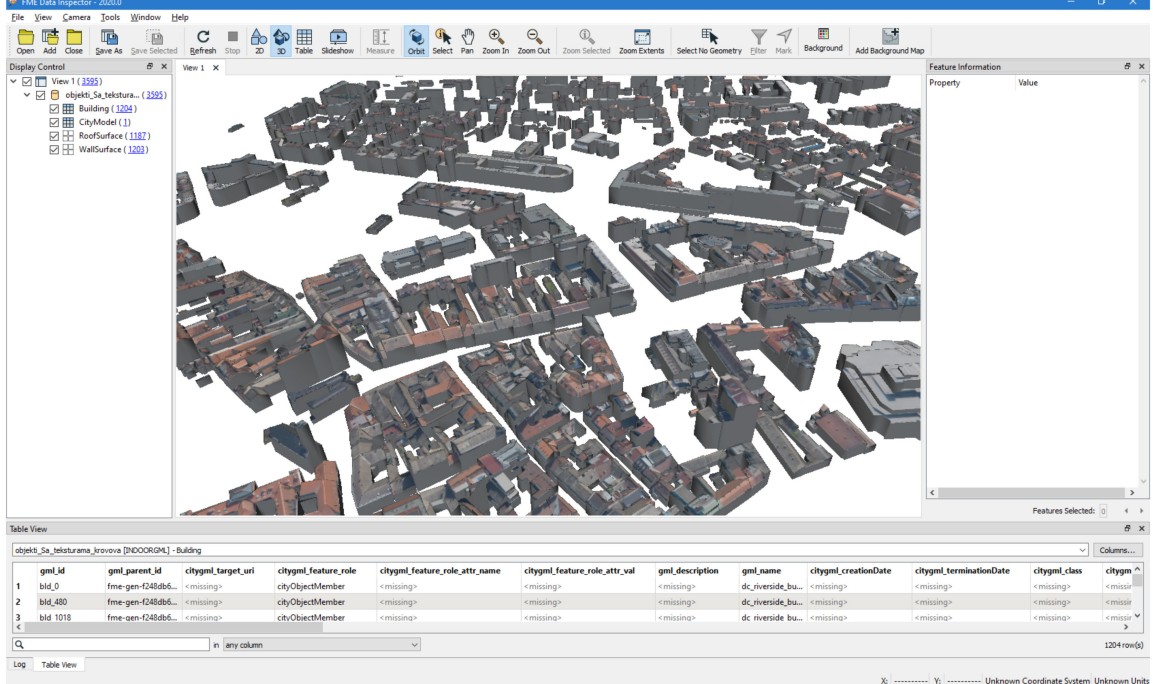

**Figure 12.** View of the 3D model with the corresponding features in Table 2 in accordance with the CitiGML standard displayed in FME Data Inspector.

CityGML is an attractive solution for hosting and sharing 3D models of cities because it combines geometry and semantics in one comprehensive data model. However, the effective visualization of 3D geometry and semantic information housed in the CityGML format is overly complex. There are several desktop applications that can visualize CityGML on local computers and some of them are FME Data Inspector, QGIS, FZK Viewer, Azul, Aristoteles, Tridicon City Discoverer Light, and SuperMap Desktop. While on the other hand, visualizing the CityGML model on the web is still a challenging area and there are not many solutions. These solutions are mainly reduced to the conversion of the CityGML model into a data model that is more suitable for displaying 3D scenes (glTF, COLLADA, X3D, OBJ, KML, and others) and during such conversions the data on attributes related to 3D models of cities are lost. Figure 12 shows the generated 3D model of the city in CityGML format with the help of FME Data Inspector.

For the purposes of visualizing the CityGML model on the web, the 3D city model created and stored in the 3DCityDB Database was then exported in glTF format to be visualized with the help of Cesium JS Web Map Client, which is optimized to display 3D city models together with all its essential features. Figures 11 and 13 show exactly 3D city model first created in accordance with the CityGML standard and then transformed into glTF format visualized with the Cesium JS Web Map Client (Philadelphia, PA 19107, USA, https://cesium.com/cesiumjs/).

The advantages of this approach to creating a 3D city model are reflected in the high degree of automation of the process of transforming the input data obtained by laser scanning into a functional

comprehensive 3D city model. This means that the required time is significantly reduced compared to the previous approach, since manual data processing and 3D modelling require a large amount of resources, both human and computer. It also reduces the impact of human errors that can occur by using methods for manual data processing and model creation. The problem that arises is that there is currently no way of texturing 3D objects completely (adding textures to the walls), and this is also a further direction of exploration or a goal to be pursued. At the time of defining methods for fully automatic texturing of objects, a comprehensive framework for the creation of 3D city models will be provided, which will contain all the steps: data collection, processing, transformation, modelling, and publication of 3D models through Internet technologies.

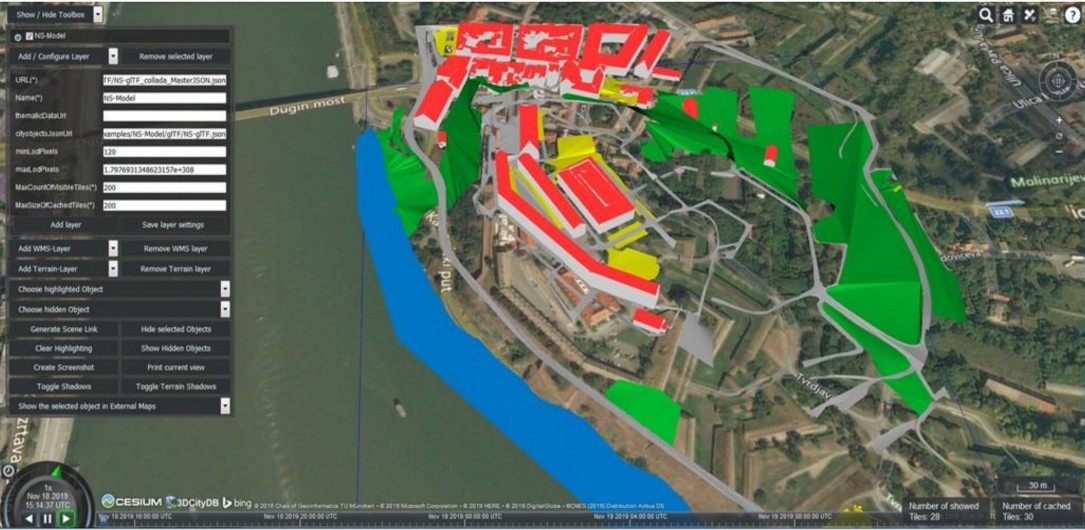

**Figure 13.** View of the 3D model with the corresponding features in Table 2 in accordance with the CitiGML standard displayed in the Cesium JS Web Map Client.

## 4.2. Data Accuracy Assement

Creating 3D building models in large scale is becoming more popular, and most popular standard of 3D models is an application schema referred to as CityGML [62]. With the usage of this standard, all city landscape elements can be modelled. In CityGML, the levels of details (LOD) are dedicated to building models [23]. These levels can be used in various variants and because of considerable interest in the standard for building models, they are still being gradually increased [63].

Today, different approaches and models of accuracy assessment of 3D building can be found [64–66]. The first group of approaches consists in comparing the created building model to the reference model, which is presented in the same form [67,68]. The second group consists in carrying out the accuracy assessment with the usage of LIDAR point clouds as reference data, where is suggested a method in which one of the crucial elements is the calculation of a distance between the 3D building model and the point cloud [69].

During LiDAR acquisition in our case study, all requirements, guidelines, and recommendation about quality assessment and quality control, from the American Society for Photogrammetry and Remote Sensing (ASPRS) were fulfilled [70]. To ensure accuracy, first activity was about calibration of the system, which is reflected in calibration flight which was performed over the control area. Vertical accuracy control was performed on nine grids of control points and the determined RMSE was below 3 cm. Horizontal accuracy control show similar results, with determined RMSE below 10 cm (Figure 14).

Outlier detection in LiDAR point clouds is also necessary process before 3D modelling, especially when 3D models are complex. So far, many studies have been done in order to remove the outliers from LiDAR data. [71,72]

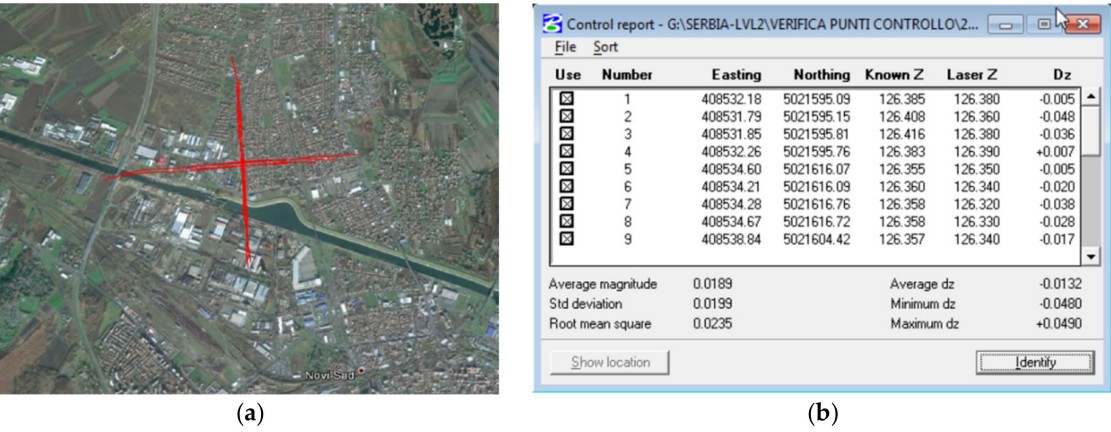

(**a**)                                                       (**b**)

**Figure 14.** Calibration flight over Novi Sad (**a**), ground control report (**b**).

In this article concept of generating objects from the LIDAR at the level of house, building, auxiliary object, residential and commercial building, was presented only at the level of CAD files, and CityGML file contain only Building, RoofSurface, and WallSurface. For this case study, the possible presence of outliers was solved during initial classification. During initial classification outliers such as low and isolated points were corrected automatically. Most of the objects in the case study area, have regular flat shapes and surfaces, so the appearance of outliers on them was minimal and was eliminated almost completely automatically by using certain TerraScan functionalities. For objects that were complex in their shape and surfaces, for complex situations such as buildings with different oriented roofs with lot of details and especially objects located closed to high vegetations (trees), it was necessary to perform manual control and correction and classification of LiDAR points. This was very important step, because these outliers can cause inaccuracies in the results obtained during 3D buildings modelling. Thus, manual classification resolves the issue at incomplete and "noisy" building-vegetation places.

## 5. Discussion and Future Developments

The focus of CityGML is on the semantical aspects of 3D city models and its structure, which allows users to employ virtual 3D city models for advanced analyses and visualizations in a variety of application domains such as urban planning, cadaster, indoor and outdoor navigation, environmental simulations, cultural heritage, etc. This contrasts with purely geometrical and graphical models such as KML, VRML, OBJ, GLTF, COLLADA, or X3D, which do not provide enough semantics. 3D city modelling of Istanbul [56], for example, used LiDAR data collected from different platforms (helicopters, car, and backpack system). They have created 3D model of buildings only in 3D CAD (DGN and DWG files) and BLOB.xml format. The BLOB files are then utilized to create LoD2 and LoD3 buildings with FME. On the other hand, CityGML is based on GML, which provides a standardized geometry and topology model of geographical features and can be combined with other geospatial data. Because of this model and its well-defined semantics and structure, CityGML can facilitate interoperable data exchange in the context of geospatial web services and can provide bases for the spatial data infrastructure for Smart City applications. We analyzed how this can be achieved in the field of 3D cadaster and urbanism, but it could be also used in other areas, such as property taxation that depends on the location (zone) of the property and its quality parameters (building year, condition of the property, whether it has central heating, exposure to the noise . . . ), modelling energy efficiency of certain city areas (building clusters), resolving issue of insufficient parking space or green areas, possibility of consumption of solar energy, etc. Therefore, the aim was to present a workflow verified on a selected case study for the development of the semantic 3D city database from LiDAR point clouds that can serve various purposes. The basic criteria for selecting activities in the workflow is that the workflow can relatively easily and efficiently be implemented into practice. For that purpose,

using state-of-the-art available tools is essential. However, despite evident advantages of the proposed workflow that tends to automate the process as much as possible, some manual work still remains, and it needs to be addressed in the future work.

*5.1. Virtual 3D City Model and Urban Planning*

Spatial planning deals with organization and regulation of space in order to improve the quality of life of its inhabitants [73]. Spatial planning is an interdisciplinary field of science and skill that uses an integrated approach considering the concept of sustainable development. The goal of an integrated approach is to link different thematic fields, whose interests overlap in certain geographic areas. In that sense, spatial data play a key role. In order to achieve the quality of the spatial plans it is necessary to analyze certain area from different aspects. Primary thematic fields that should be taken into account in the spatial planning are: urban and rural settlements, industry, infrastructure, public services, natural resources, environment and cultural heritage. Disciplines that involve spatial planning include land use planning, urban planning, regional planning, transport planning, and environmental planning. Spatial planning can be done on local, regional, national, and international levels and result in the creation of spatial plans.

The Serbian Law on Planning and Construction [74] defines the spatial plan as a planning document. It defines several types of plans including: the spatial plan of the Republic of Serbia, the regional spatial plans, the spatial plans of local communities (municipalities), the spatial plans for the special-purpose area, and the urban plans.

There are different kinds of public companies in Serbia that are related to the urban and spatial planning of municipalities or cities. Main objective is to perform continuous work on spatial and urban planning of the environment and settlements of strategic importance for the Republic of Serbia. Main focus of public companies on a municipality level is related to development of the spatial plan of the city and the spatial plans of the municipalities, development of spatial plans for special purpose areas, development of urban plans for the territory of the city, decision-making on drafting planning documents, development of urban projects, creation of detailed regulation plans, projects for land allotment, strategic assessment projects, building requirements, and other professional activities in the field of urban and spatial planning and environment.

Spatial data in these companies is mainly provided by means of georeferenced spatial plans with different details, which depends on the purpose of those plans. Common feature of such plans is that they are usually created using CAD tools in a local coordinate system, together with textual explanation and they contain many layers (often more than a hundred). Problems that are identified during this process of creation are summarized by Sladić et al. [75]. One of the issues identified is impossibility of global perception of space, which can be improved by utilizing 3D spatial data and 3D city models. Although 3D spatial data has been used in urban planning in Novi Sad, semantic models such as CityGML were not used.

According to the previously explained example from cadaster, 3D city models in CityGML format can be used to enhance different 2D or 3D vector layers, that represent layers in general and detailed regulation plans (obtained relatively quickly by mass LiDAR surveying). In order to be related to the current state in different urbanism spatial detailed regulation plans of the city of Novi Sad, CityGML, or IFC datasets should be linked to the databases that contain alphanumeric data about land use and land cover such as: water surface, green area, embankment, park area, sport area, pavement, cemetery, cycle path, sidewalk, open channel in the settlement, power line, tall trees, grass, hardwood, conifers, etc. However, those layers are different from city to city, and in Serbia a unique and standardized number of these layers defined by the law does not exist. So, this example is just for the city of Novi Sad (second largest city in Serbia) and for public company JP Urbanism of the city of Novi Sad.

As a previously explained, CityGML standard has already anticipated a mechanism to link an object from a 3D model to a corresponding object in another information system, through external references. This reference consists of the name of the external information system, represented by a

Uniform Resource Identifier (URI), which is a generic format for references to any kind of resources on the Internet, and the reference of the external object, given either by a string (name) or by an URI. Table 3 shows an external reference that contains URI to information system of the detailed spatial plans of the current state for the spatial public companies of the city of Novi Sad, and in this case it is represented by the unique id of the land use and land cover type. Other names of the features that can be pointed to by the external reference may be: 603-p-low voltage power line, 209-p-railway, 204-p-sidewalk, 201-p-road, 514-p-open channel in the settlement, 136-p-public cemetery, 403-p-canopy, 601-p- transmission line . . .

**Table 3.** External reference of the land use and land cover to a detailed spatial plans database.

| External Reference | Name | URI |
|---|---|---|
| | informationSystem | http://www.nsurbanizam.rs/pgr |
| | externalObject.name | 142-p-public green area |

One of the important areas of application of the developed CityGML-based 3D city model related to urban planning is the development of strategic noise maps. In 2002, the European Union adopted the environmental noise assessment and management directive [76], which requires the development of strategic noise maps and action plans in order to reduce environmental noise. This directive has been implemented in the Republic of Serbia in the Law on Protection against Environmental Noise [77]. A strategic noise map is a map that presents data on noise levels in an area and is used to estimate the total noise exposure of a particular area from different noise sources or to predict the total noise in an area. The objective of strategic noise maps is to clearly present noise status to both authorities and citizens, so that they can work together to reduce the number of people exposed to unacceptable levels of noise. Environmental noise is measured every month on a network of 16 measurement sites conducted by the Institute of Public Health of Vojvodina, Novi Sad and it includes measuring a set of parameters such as average traffic flow. Based on these parameters an estimate of the noise level is than calculated.

Strategic noise maps are produced using computational methods and programs for calculating the emission and propagation of noise emitted by known noise sources in the geographical space of known characteristics. The methodology for the preparation of strategic noise maps is defined in the Rulebook on the manner of preparation and content of strategic noise maps and how they are presented to the public [78] and based on the guidelines given in European Directive.

The basic requirement is that the data necessary for the development of strategic noise maps should be such that they enable their easy implementation in the city GIS database and in the software intended for the development of strategic noise maps. The necessary data for the development of strategic noise maps are 3D digital terrain model, 3D model of buildings with associated purpose of the buildings, noise model caused by traffic, and noise model caused by industrial sources.

Public company JP Urbanism of the city of Novi Sad conducted the project for the development of strategic noise map as a legal obligation [79]. Figure 15a shows the 3D model used for the development of the strategic noise map. This 3D model was developed based on 2D data obtained from the city GIS database of the building footprints and heights of the buildings that were measured during the field survey. For those measurements it took four employees and 50 working days to cover the area defined by the project (which included university campus and residential area called Liman). On the other side Figure 15b shows the same area in CityGML produced by the previously described methods.

The conclusions and recommendations during the development of strategic noise map is to use aerial surveying techniques that will allow recording of the heights of the buildings and more detailed recording of the terrain configuration, which will greatly speed up the development of the strategic noise map. In this sense, a proposed methodology based on LiDAR surveying and virtual 3D city models can greatly help. Furthermore, strategic noise map should be regularly updated and aligned with changes in urban environment and it is necessary to take measures where noise limit values are

exceeded. Also, real time noise measuring by IoT devices can be introduced to substitute subjective observations obtained through interviewing the citizens about the noise level.

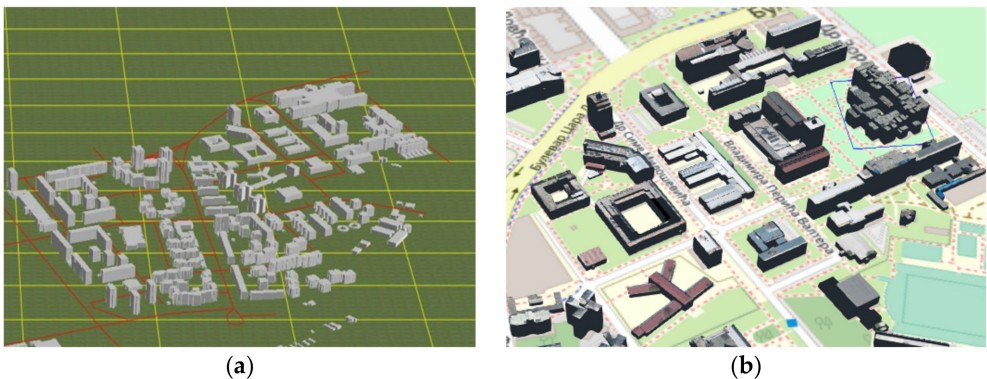

| (**a**) | (**b**) |

**Figure 15.** Used 3D model (**a**) [56], CityGML based model (**b**).

## 5.2. Virtual 3D City Model and 3D Cadastre

Usage of 3D semantic formats such as CityGML and IFC for the purpose of 3D cadaster in Serbia has already been analyzed. Radulović et al. [80] analyzed the possibility of generating CityGML-based 3D cadaster based on 2D footprints and floor plans and other attribute data that is stored in cadastral offices. IFC based 3D cadaster in Serbia for buildings and building units was proposed by Sladić et al. [75]. Such data produced by architects and technical drawers can be reused for the purpose of 3D cadaster, where it is most effective for the representation of buildings and their building units (condominiums, business spaces and garages). However, 3D cadaster in Serbia based on data collected from LiDAR surveying has not been proposed yet. The main advantage of such data collection method is that it can collect and produce mass data but without the indoor space.

The organization of real estate cadaster and proposed LADM profile for Serbia are described by Radulović et al. [80] and consists of two types of datasets: alphanumeric legal data about rights on properties and spatial data. Spatial data in Serbian cadaster is provided by means of georeferenced cadastral maps that contain 2D boundaries of parcels and buildings, while the spatial data about building units is kept in the form of sketches i.e., 2D floor plans (usually created using CAD tool in the local coordinate system). With this analogy, 3D city models in CityGML format can be used as an enhanced 3D cadastral map of buildings (obtained relatively quickly with mass LiDAR surveying) on top of 2D vector layer that represents parcels, while indoor legal space (building units) may be represented using CityGML LOD4, IndoorGML, or IFC. In order to be related to 3D cadaster, CityGML and IndoorGML or IFC datasets should be linked to the databases that contain alphanumeric data about properties, property rights, and right holders. One way to achieve this is to add attributes to the appropriate legal spaces that uniquely define one legal space (property).

CityGML standard has already anticipated a mechanism to link an object from a 3D model to a corresponding object in another information system, such as cadastral information system. This is called an external reference represented by ExternalReference class. If the informationSystem element is missing in the ExternalReference, the ExternalObjectReference must be an URI. The reference of a 3D object to its corresponding object in an external data set is essential, if an update must be propagated or if additional data are required, for example the name or address of a building's owner in a cadastral information system. In order to provide such information, each CityGML object may refer to external data sets using the concept of ExternalReference. The generic concept of external references allows for any CityGML object an arbitrary number of links to corresponding objects in external information systems. Table 4 shows an external reference of the building that contains URI to cadastral information system and the name of the external object which in this case is represented by the unique property identification number.

**Table 4.** External reference of the building to a cadastral database.

| External Reference | Name | URI |
|---|---|---|
| | informationSystem | https://katastar.rgz.gov.rs/eKatastarPublic |
| | externalObject.name | 89010-1124-0-1-1 |

Official cadastral maps in Serbia contain 2D spatial data in vector format (CAD and GIS) of the boundaries (footprints) of the parcels and buildings. To analyze the consistency of such data with the obtained 3D objects in CityGML it is necessary to overlay 3D data with the official 2D spatial data of the boundaries. Geometric inconsistencies can be solved by georeferencing according to the official cadastral map. There is also a possibility (although relatively rare) that a LiDAR survey may reveal objects that are not appropriately stored in the official cadastral data sets (e.g., missing or changed). This will be revealed through linking objects to cadastral database and each case should be solved individually through the official cadastral procedures. Mapping agency in Serbia already use satellite imagery to detect illegal buildings that are built without building permit and are not recorded in the real estate cadaster, but it could also benefit from this method. LiDAR surveying may also be used to update data in the real estate cadaster.

## 6. Conclusions

In this paper we demonstrated the process from data collection by LiDAR surveying, through ETL transformations and data processing to developing 3D virtual city model. The results are presented on the case study of Campus area of the University of Novi Sad, nearby Petrovaradin Fortress and part of the city center. The methodology proposed in this paper resulted in 1204 models of buildings developed and stored in a database in compliance with the CityGML standard. Total of 12% of all the models were houses, 3% were auxiliary objects, 3% were residential and commercial buildings, and the biggest share were buildings, 82%. We discussed how this type of modelling can be applied in 3D cadaster and modern urban planning. Future work is planned in two directions. One is the improvement of the 3D city modelling process to include automatization of the process where possible, to enhance photorealistic visualization, to fully develop 3D city database and populate it with relevant attributes and link it with already existing relevant information systems, etc. The other direction is to expand areas of its potential applications and to provide overall 3D spatial data infrastructure for future Smart Cities and IoT-based applications on a city-scale level to improve city services in response to the needs of its citizens and their wellbeing and quality of life. This will include more thorough analysis of the application of ICT and IoT in Smart City context. The first part includes the development of machine learning-based algorithms to significantly decrease the time between data acquisition and its application, intending to achieve real-time or near real-time levels. The second part refers to the development of 3D services capable of data dissemination through web-based applications and their specialization in particular areas.

**Author Contributions:** Conceptualization, Dušan Jovanović, Miro Govedarica; methodology, Dušan Jovanović, Miro Govedarica; investigation, Stevan Milovanov, Igor Ruskovski, Vladimir Pajić; supervision, Dušan Jovanović, Miro Govedarica; data curation, Vladimir Pajić; writing—original draft, Dušan Jovanović, Stevan Milovanov, Igor Ruskovski, Dubravka Sladić; writing—review and editing, Dubravka Sladić, Aleksandra Radulović. All authors have read and agreed to the published version of the manuscript.

**Funding:** This research received no external funding.

**Acknowledgments:** Results presented in this paper are part of the research conducted within the Grant No. 37017, Ministry of Education and Science of the Republic of Serbia.

**Conflicts of Interest:** The authors declare no conflict of interest.

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
