# Peer review of "Building Virtual 3D City Model for Smart Cities Applications: A Case Study on Campus Area of the University of Novi Sad"

_ijgi, doi:10.3390/ijgi9080476_

Round 1
Reviewer 1 Report
Paper is well structure. the background and bibliography are well defined. Aims, workflow and result are clear.
The only suggestion is to pay attention to the size of the text in the images that describe the pipeline. Figure 06 text is not readable. Fig 8 and Fig9 text also should be enlarged.
Reviewer 2 Report
The article "Building Virtual 3D City Model for Smart Cities Applications: A Case Study on Campus Area of the University of Novi Sad" presents a very interesting issue related to the development of virtual 3D city model based on airborne LiDAR surveying and to analyse its applicability towards Smart Cities applications. The work is focused on the case study of campus area of the University of Novi Sad but the methodology and the adopted course of action may be universal, which increases the value of the article.
The title was formulated correctly and closely refers to the content of the article. The content of the article is very clear, readable and correctly divided. I really appreciated that the authors present the future developments of this methodology and how it can be achieved in various fields. However, in my opinion the content of the Conclusion needs to be slightly improved. Moreover, the literature review about the information and communication technology (ICT) and Internet of Things (IoT) related to the Smart City is modest.
After implementing these minor revisions, as far as I'm concerned the article can be accepted.
Reviewer 3 Report
- Many algorithms have been used in this Paper without clarifying about its reference and efficiency ( e.g.:
- 256 : quick automatic algorithms
- 270: digital terrain model derived from the cloud points
- 271: filtering only the significant points of the ground
- 277 : basic classification
- 278 : creation of digital elevation model
- …
- The first ETL transformation scheme for vectorized CAD models into GML model without textures (this process is Unclear and understandable)
- Figure 3 should be approve
- there is no spatial resolution analysis in this article.
Reviewer 4 Report
The thematic focus of the work has a current topic. Development of 3D Smart citiy is a current topic and its solution is essential for the society. Addressing this issue is necessary at all levels involved, which confirms the content structure of the paper. The authors presented a comprehensive 3D model of the city with regard to the standards adopted in the implementation of 2D / 3D objects in the environment of virtual cities. The current approach in the field of 3D data collection, exchange formats, standardization and their visualization was taken into account.
Recommendations for addition:
- Recommend the authors to include in the Introduction section a brief overview of spatial data quality, given that the quality of spatial data has a major impact on the creation of a 3D model of the city. I recommend in the section Introduction to emphasize the topicality of the solved issue by study and supplementing the overview of other professional works that deal with the given issue. To avoid having to search for a long time, I offer an overview of a few posts on this issue: https://doi.org/10.3390/ijgi9010016, https://doi.org/10.3390/app10030820.
- Did the authors consider by the concept of generating objects from the LIDAR at the level of House, Building, Residential and commercial building, even with the possible presence of located outliers? Especially for buildings, these outliers can cause inaccuracies in the results obtained.
- In section 5. Discussion and future developments, is needed a deeper discussion to evaluate and compare the achieved results with other scientific works of given importance to emphasize the successful implementation of the presented proposal in practice.
- Section 5.2 Virtual 3D city model and 3D cadastre, how was solved in the 3D cadastre the inconsistency of data necessary for the creation of this environment.
Round 2
Reviewer 3 Report
The introduction is too long .. it should be shortened..
It should not explain everything to the reader in the introduction..
